# Detection of Cancer Recurrence Using Systemic Inflammatory Markers and Machine Learning after Concurrent Chemoradiotherapy for Head and Neck Cancers

**DOI:** 10.3390/cancers15143540

**Published:** 2023-07-08

**Authors:** Yoon Kyoung So, Zero Kim, Taek Yoon Cheong, Myung Jin Chung, Chung-Hwan Baek, Young-Ik Son, Jungirl Seok, Yuh-Seog Jung, Myung-Ju Ahn, Yong Chan Ahn, Dongryul Oh, Baek Hwan Cho, Man Ki Chung

**Affiliations:** 1Department of Otorhinolaryngology-Head & Neck Surgery, Inje University College of Medicine, Ilsan Paik Hospital, Goyang-Si 10380, Republic of Korea; hn0131@paik.ac.kr (Y.K.S.); cty896@gmail.com (T.Y.C.); 2Medical AI Research Center, Samsung Medical Center, Seoul 06351, Republic of Korea; zero.kim@skku.edu (Z.K.); mjchung@skku.edu (M.J.C.); 3Department of Data Convergence and Future Medicine, Sungkyunkwan University School of Medicine, Seoul 06351, Republic of Korea; 4Department of Otolaryngology-Head & Neck Surgery, Sungkyunkwan University School of Medicine, Samsung Medical Center, Seoul 06351, Republic of Korea; chbaek@skku.edu (C.-H.B.); yison@skku.edu (Y.-I.S.); 5Center for Thyroid Cancer, Department of Otolaryngology-Head and Neck Surgery, Research Institute and Hospital, National Cancer Center, Goyang-si 10408, Republic of Korea; junn279@gmail.com (J.S.); jysorl@ncc.re.kr (Y.-S.J.); 6Divison of Hematology and Medical Oncology, Department of Medicine, Sungkyunkwan University School of Medicine, Samsung Medical Center, Seoul 06351, Republic of Korea; silkahn@skku.edu; 7Department of Radiation Oncology, Sungkyunkwan University School of Medicine, Samsung Medical Center, Seoul 06351, Republic of Korea; ahnyc@skku.edu (Y.C.A.); dongryul.oh@samsung.com (D.O.); 8Department of Biomedical Informatics, CHA University School of Medicine, CHA University, Seongnam-Si 13488, Republic of Korea; 9Institute of Biomedical Informatics, School of Medicine, CHA University, Seongnam-Si 13488, Republic of Korea

**Keywords:** deep neural network, machine learning, head and neck cancers, cancer recurrence, concurrent chemoradiotherapy

## Abstract

**Simple Summary:**

The neutrophil-to-lymphocyte ratio (NLR) and the platelet-to-lymphocyte ratio (PLR) are known as prognosticators in head and neck cancers, but the clinical significance of those values after treatment remains unclear. In this study, we aimed to investigate the relationship between temporal changes in the NLR and PLR values after treatment and cancer recurrence in head and neck cancer patients. We proposed developing a DNN model that utilizes these changes to predict recurrence after CCRT, comparing its performance with traditional machine learning (ML) models. The results of this study showed that the DNN model outperformed the other models, and feature selection using the recursive feature elimination algorithm further improved the performance, with ROC-AUC and PR-AUC values of 0.883 ± 0.027 and 0.778 ± 0.042, respectively. The results could improve prognostication and risk assessment in head and neck cancers. Additionally, this study may inspire further research and advancements in utilizing advanced ML techniques for personalized cancer management and surveillance.

**Abstract:**

Pretreatment values of the neutrophil-to-lymphocyte ratio (NLR) and the platelet-to-lymphocyte ratio (PLR) are well-established prognosticators in various cancers, including head and neck cancers. However, there are no studies on whether temporal changes in the NLR and PLR values after treatment are related to the development of recurrence. Therefore, in this study, we aimed to develop a deep neural network (DNN) model to discern cancer recurrence from temporal NLR and PLR values during follow-up after concurrent chemoradiotherapy (CCRT) and to evaluate the model’s performance compared with conventional machine learning (ML) models. Along with conventional ML models such as logistic regression (LR), random forest (RF), and gradient boosting (GB), the DNN model to discern recurrences was trained using a dataset of 778 consecutive patients with primary head and neck cancers who received CCRT. There were 16 input features used, including 12 laboratory values related to the NLR and the PLR. Along with the original training dataset (*N* = 778), data were augmented to split the training dataset (*N* = 900). The model performance was measured using ROC-AUC and PR-AUC values. External validation was performed using a dataset of 173 patients from an unrelated external institution. The ROC-AUC and PR-AUC values of the DNN model were 0.828 ± 0.032 and 0.663 ± 0.069, respectively, in the original training dataset, which were higher than the ROC-AUC and PR-AUC values of the LR, RF, and GB models in the original training dataset. With the recursive feature elimination (RFE) algorithm, five input features were selected. The ROC-AUC and PR-AUC values of the DNN-RFE model were higher than those of the original DNN model (0.883 ± 0.027 and 0.778 ± 0.042, respectively). The ROC-AUC and PR-AUC values of the DNN-RFE model trained with a split dataset were 0.889 ± 0.032 and 0.771 ± 0.044, respectively. In the external validation, the ROC-AUC values of the DNN-RFE model trained with the original dataset and the same model trained with the split dataset were 0.710 and 0.784, respectively. The DNN model with feature selection using the RFE algorithm showed the best performance among the ML models to discern a recurrence after CCRT in patients with head and neck cancers. Data augmentation by splitting training data was helpful for model performance. The performance of the DNN-RFE model was also validated with an external dataset.

## 1. Introduction

Systemic inflammation is known to suppress the anticancer immune response [1,2], and systemic inflammatory markers have been associated with poor prognosis for various cancers [3,4,5,6,7,8,9,10]. Neutrophilia and thrombocytosis have been shown to reflect tumor progression. In addition, lymphopenia, which indicates a compromised antitumor immune response, has been associated with a poor prognosis in various advanced solid tumors. The neutrophil-to-lymphocyte ratio (NLR) and the platelet-to-lymphocyte ratio (PLR) may capture the combined effects of these opposing factors. High NLR and PLR values have been widely recognized as poor prognostic indicators for recurrence and survival in head and neck cancers [3,11,12,13,14]. However, it has not been studied whether temporal changes in the NLR and PLR values after treatment are related to the development of recurrence. Post treatment, patients are followed up continuously for cancer recurrence, and blood tests are performed along with imaging studies. Resultantly, large amounts of laboratory data have been accumulated. Apart from the pretreatment values of the NLR and PLR, the temporally collected values of NLR and PLR after the end of the treatment may have separate clinical significance. They can be used to detect recurrence during the follow-up of cancer patients, and, considering the vast amount of the data, they are also suitable for use in machine learning (ML) or deep learning (DL).

ML is a collection of methods that allow software applications to develop algorithms to predict outcomes from existing datasets without being explicitly programmed. DL is an advanced subset of machine learning based on artificial neural networks with representation learning. Deep neural network (DNNs), which are representative architectures of DL, have recently made significant progress in developing predictive models in the medical field using electronic medical record (EMR) datasets [15,16,17]. It is known that DNNs are superior to conventional ML models in handling vast amounts of data.

In this study, we aimed to develop a DNN model to discern cancer recurrence from the temporal NLR and PLR values during follow-up after chemoradiation and to evaluate the performance of the DNN model compared with conventional ML models.

## 2. Materials and Methods

### 2.1. Patients

This study was conducted on a cohort of primary head and neck cancer patients who had received definitive concurrent chemoradiotherapy (CCRT) from March 1994 to March 2019. Initially, 1212 consecutive patients were screened for enrollment, and among them, patients were excluded with other diseases that could affect blood test values, for example, synchronous or metachronous malignancies, hematologic diseases such as idiopathic thrombocytopenic purpura, HIV infection, liver cirrhosis, and active tuberculosis. We also excluded patients with missing analytical items, including missing blood test data, missing medical records on recurrences, and missing follow-up status, as well as patients who smoked. Patients were excluded from this study who developed recurrence within three months after completion of CCRT, which was considered to be a remnant disease. Lastly, those who were lost to follow-up within three months were also excluded. Finally, 758 patients were included. This study was approved by the Institutional Review Board of Samsung Medical Center (SMC IRB file no. 2020-06-125-001) and was conducted following the Helsinki Declaration. Informed consent was waived because of the retrospective design.

### 2.2. Training Dataset and Input Features

This study used the EMR data of 758 primary head and neck cancer patients who had received definitive concurrent chemoradiotherapy (CCRT) from March 1994 to March 2019. The original dataset was retrieved via a retrospective review of the patients’ EMRs. The outcome label is a binary variable indicating the development of recurrence. Recurrence was decided based on clinical, radiological, or pathological descriptions from EMRs and included regional, local, and distant recurrences. We utilized 16 input features for training the ML models, which included each patient’s age, sex, smoking status, stage (based on the AJCC 7th system), and 12 blood laboratory features. Laboratory values up to the diagnosis of recurrence were used for recurrent cases, and laboratory values up to the end of the follow-up with a 5-year limit were used for non-recurrent cases. Recent deep learning technologies have the capability to capture the temporal patterns of time series data; however, it is often very difficult to learn such a pattern if the amount of data is limited and data have been acquired irregularly for each patient. In addition, the duration of follow-ups differed from each other, and the visits were also often irregular. Therefore, we extracted 12 features from the irregular longitudinal blood laboratory values for each case (Table 1) [18]. The NLR and PLR values were calculated by dividing the neutrophil count and the platelet count by the lymphocyte count, respectively. For the NLR, the following six blood laboratory features were calculated during the corresponding period in recurrent patients: mean NLR (NLR _mean_), maximum–minimum NLR (NLR _max–min_), NLR after treatment (NLR _Tx_), NLR at the time of recurrence (NLR _R_), the difference between NLR _R_ and NLR _mean_ (NLR _R-mean_), and the difference between NLR _R_ and NLR _Tx_ (NLR _R-Tx_). The NLR _Tx_ and PLR _Tx_ were obtained from 14 days to 90 days after cessation of CCRT. The NLR _R_ and PLR _R_ were obtained from 90 days before the detection of recurrence to 30 days after the detection of recurrence. In non-recurrent patients, the NLR _R_ means the NLR at the end of the period. For the PLR, six blood test indices were extracted in the same manner.

Missing values were removed, and categorical variables were encoded using the one-hot encoding technique since there was no ordinal relationship. Subsequently, the 12 extracted input variables were normalized to a standard scale. Additionally, recursive feature elimination was performed to assess the impact of each input variable on the response variable. The contribution of each variable to the prediction was calculated as a discrete interval encompassing all the variables in the training dataset. The response probability was estimated for each range of the input variable, while keeping the other variables at their average value for numeric variables or at zero for binary variables. In each iteration, the input variable with the smallest dynamic range of the output response was successively removed until only one variable remained.

### 2.3. Split Training Dataset

In order to overcome the limited size of the training data, we tried to adopt data augmentation by data splitting. For each recurrent case, the temporal blood laboratory values were split into two groups based on 180 days before the diagnosis of recurrence (Figure 1a). Values more than 180 days before recurrence were reclassified as the values of the non-recurrent state. We hypothesized that data until 180 days before recurrence could be treated as negative (non-recurrent) samples, which could be helpful for better model performance. In that way, a modified dataset was also built (Figure 1b).

### 2.4. Machine Learning Models

For both the original and the modified dataset, traditional ML models and a DNN model were built. Among the traditional supervised ML algorithms, random forest (RF), logistic regression (LR), and gradient boosting (GB) were used. On the one hand, logistic regression is a statistical model used for binary classification problems. It estimates the probability of an instance belonging to a particular class by utilizing a logistic function. This model assumes a linear relationship between the input features and the log-odds of the target class; however, it may struggle to capture complex nonlinear relationships and may not perform as effectively as more advanced models in highly complex tasks.

Random forest, on the other hand, is an ensemble learning method that combines multiple decision trees to make predictions. It creates a forest of decision trees, where each tree is trained on a random subset of the data and features. Random forest is capable of performing both classification and regression tasks and can handle complex interactions and nonlinear relationships in the data; however, it can be prone to overfitting if the number of trees is too high and may not perform as well on high-dimensional datasets.

Gradient boosting is another ensemble learning technique that builds an ensemble of weak prediction models, typically decision trees, in a sequential manner. Gradient boosting combines the predictions of multiple weak models to make a final prediction. It is capable of handling complex relationships, capturing interactions, and performing well on structured and tabular data; however, it can be computationally expensive and requires careful tuning of hyperparameters.

In contrast to logistic regression, which assumes a linear relationship between the input features and the log-odds of the target class, deep neural networks (DNNs) can approximate any complex function by learning hierarchical representations through hidden layers. This flexibility enables DNNs to handle highly nonlinear data and to capture intricate interactions and dependencies between features. DNNs excel in tasks where the relationships in the data are complex and may not be easily captured by simpler models.

To develop the ML models, the entire dataset was divided into 10 folds by applying stratified random splits. The model was trained with nine of the folds, and the performance of the model was measured repeatedly with the remaining fold. The performance of the model was measured using a receiver operating curve and the area under the curve (ROC-AUC) and precision recall and the area under the curve (PR-AUC) values. The models were developed on Ubuntu Environment using Python 3.5. The DNN algorithms were developed, including all input features or selected features using the RFE method [19]. In the DNN-RFE model, the weakest features were removed until the ROC-AUC and PR-AUC values reached their peaks. The hyperparameters of the DNN models were tuned to reach the highest mean of the ROC-AUC and PR-AUC values. The number of units, batch size, learning rate, and dropout rate were tuned. Adam was used as an optimizer, and the optimal learning rate was 0.001 [20].

### 2.5. External Validation Dataset

The validation cohort was enrolled separately in an unrelated institution. A total of 173 primary head and neck cancer patients who had received definitive concurrent chemoradiotherapy (CCRT) from April 2002 to March 2019 were enrolled for validation. This validation dataset was also built in two versions, i.e., the original validation dataset and a modified validation dataset (split dataset).

## 3. Results

### 3.1. Patients’ Characteristics

The clinicopathological profiles of 778 patients included in the training dataset study are presented in Table 2. The mean age of the patients was 60.4 years (standard deviation, 13.0 years). There were 625 males and 153 females. The two most common tumor sites were the nasopharynx (38.0%) and the oropharynx (29.3%). Pathologically, squamous cell carcinoma (64.7%) and undifferentiated carcinoma (31.2%) accounted for most cases. Approximately 90% of patients were in advanced stages. In total, there were 489 cases (62.9%) in stage IV and 209 cases (26.9%) in stage III. Among all enrolled cases, 206 (26.5%) cases had a recurrence, 108 cases had locoregional recurrences without distant metastasis, and 98 cases had distant metastasis with or without locoregional recurrence. The five-year recurrence-free survival (RFS) rate was 66.5%, and the median RFS was 189.8 months.

With data splitting of the recurrent cases, which had no recurrence within 180 days after the end of the treatment, 117 non-recurrent cases were newly created (Figure 1). Resultantly, 206 recurrent cases and 689 non-recurrent cases were included in the split training dataset.

In the external validation dataset, 73 recurrent cases and 100 non-recurrent cases were included. After data splitting of recurrent cases, 73 recurrent cases and 152 non-recurrent cases were included in the split validation dataset. We tried to split the external validation dataset in order to test whether our ML models could accurately recognize the sample data, until 6 months before recurrence, as negative (non-recurrent) cases. The clinicopathological characteristics of the external validation dataset are presented in Appendix A.

### 3.2. Cross-Validation Performance on Training Datasets

The ROC-AUC values of the LR, RF, and GB models were 0.696 ± 0.059, 0.812 ± 0.049, and 0.600 ± 0.052, respectively, on the original (non-split) training dataset (Figure 2a). The ROC-AUC value of the DNN model using 16 input features was 0.828 ± 0.032 on the original training dataset (Table 3, Figure 2a). The PR-AUC values of the LR, RF, GB, and DNN models were 0.454 ± 0.079, 0.622 ± 0.081, 0.346 ± 0.055, and 0.663 ± 0.069, respectively, on the original training dataset.

When the models were trained using a split training dataset, the cross-validation performances were also measured. On the split training dataset, the ROC-AUC values of the LR, RF, GB, and DNN models were 0.700 ± 0.053, 0.751 ± 0.053, 0.735 ± 0.057, and 0.837 ± 0.038, respectively (Figure 2b). The PR-AUC values of the LR, RF, GB, and DNN models were 0.456 ± 0.072, 0.539 ± 0.084, 0.516 ± 0.085, and 0.666 ± 0.045, respectively.

When the RFE algorithm was applied to the DNN model on the original training dataset, the top three features which maximized the ROC-AUC and the PR-AUC values of the DNN model were selected (Figure 3a). The selected features were PLR _mean_, PLR _Tx_, and PLR _max–min_. With these three features, the ROC-AUC and the PR-AUC values of the DNN-RFE model were 0.884 ± 0.032 and 0.788 ± 0.032, respectively, on the original training dataset (Table 4). The DNN-RFE algorithm was applied to the split training dataset, and PLR _R_, PLR _R-Tx_, PLR _mean_, PLR _Tx_, and PLR _max–min_ features were selected (Figure 3b). With these top five features, the ROC-AUC and the PR-AUC values of the DNN-RFE model were 0.889 ± 0.032 and 0.771 ± 0.044, respectively, on the split training dataset. When these top five features were used on the original training dataset, the ROC-AUC and the PR-AUC values were 0.883 ± 0.027 and 0.778 ± 0.042, respectively (Table 4).

The DNN-RFE model trained with a split training dataset had the maximal performance, and five blood laboratory features were included in the model (PLR _R_, PLR _R-Tx_, PLR _mean_, PLR _Tx_, and PLR _max–min_).

### 3.3. External Validation (Figure 4)

We evaluated our ML models on the original external dataset (*N* = 173). Each ML and DNN model trained with the original and the split training dataset was evaluated on the original external validation dataset (*N* = 173) (Figure 4). From the original training dataset, the ROC-AUC values of the LR, RF, GB, DNN, and DNN-RFE models were 0.572, 0.748, 0.585, 0.643, and 0.710, respectively (fourth column in Table 5). When we evaluated the models trained with a split training dataset, the ROC-AUC values of the LR, RF, GB, DNN, and DNN-RFE models were 0.556, 0.531, 0.672, 0.766, and 0.784, respectively (third column in Table 5).Figure 4Overview of the external validation process for both original dataset model and split dataset model on each validation dataset.
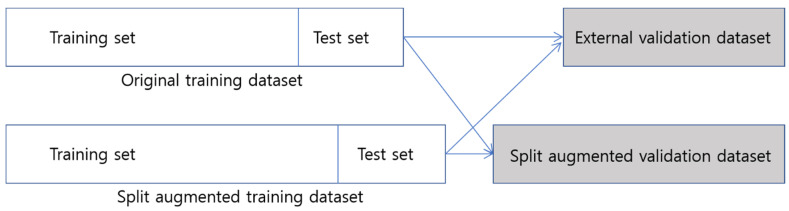


Additionally, we evaluated our ML models on a split external validation dataset again (*N* = 225) (Figure 4b). When we evaluated the split validation dataset, the ML and DNN models trained with the split training dataset showed higher performances than the models trained with the original training dataset. The ROC-AUC value of the DNN-RFE model trained with the split training dataset was higher than that of the DNN-RFE model trained with the original dataset (0.845 vs. 0.708 in Table 5).

## 4. Discussion

Inflammatory responses within the tumor microenvironment play important roles in cancer initiation and progression [21,22,23]. The systemic inflammatory response is associated with peri-tumoral or intra-tumoral inflammation. The systemic inflammatory response can suppress the antitumor activity of immune cells, such as activated T cells and natural killer cells [1,2]. Recently, systemic inflammatory markers have been widely studied as prognostic factors in various cancers, including head and neck cancers. Among several markers, the NLR and the PLR values are the most well-established prognosticators. A meta-analysis of 100 clinical studies covering a variety of solid tumors, including head and neck cancers, showed that higher NLR values were significantly associated with shorter OS with a hazard ratio (HR) of 1.81 (95% CI = 1.67 to 1.97). Higher NLR values were also associated with shorter disease-free survival (DFS) (HR, 2.27, 95% CI = 1.67 to 1.97) [24]. An elevated PLR has also been reported to be a poor prognostic factor. In a meta-analysis of 26 studies including 13,964 patients, elevated PLR values were a significant factor for worse OS with a HR of 1.60 (95% CI = 1.35–1.90) [25]. Various inflammatory markers have also been studied for head and neck cancers alone, and NLR and PLR values have been reported as independent prognostic factors [12,13,14,26].

Post-treatment NLR and PLR values have also been reported to have prognostic value in various cancers. Radiation therapy (RT) and concurrent chemoradiation therapy (CCRT) for head and neck cancers can frequently induce lymphopenia and other blood cell count changes [27,28]. The resultant increases in NLR or PLR values can have prognostic significance independent of the pretreatment NLR or PLR values [29,30,31].

However, studies on the NLR and the PLR have focused on the prognostic significance of either pretreatment or post-treatment values. Previous studies have been intended to evaluate the risk of future death or cancer recurrence in patients with presently elevated NLR or PLR values. Therefore, pretreatment or post-treatment NLR and PLR values have been classified as prognostic factors such as tumor stage. Prominent prognostic factors for cancer recurrence include stage, pathology, and site. Tumors diagnosed at an advanced stage, those exhibiting aggressive pathology with poor differentiation, or those located in sites such as the oral tongue are associated with poor prognosis. In addition to these factors, patients with higher NLR or PLR values at diagnosis are now believed to have an increased likelihood of experiencing recurrence after treatment.

Unlike previous studies, we employed serially measured NLR and PLR values after the completion of treatment, rather than point values at diagnosis. Our objective was not to assess the prognostic values of those systemic inflammatory markers. Instead, we aimed to determine whether the temporal values measured after treatment could indicate the presence of recurrence, if it occurs. Various modalities, such as sonography or CT scan, can detect cancer recurrence during the follow-up after treatment. Systemic inflammatory markers and blood test results can be used as intermediate means to detect recurrence, although not as directly as radiologic tests. As we used in this study, blood tests are continuously checked after treatment, and enormous amounts of test data can be accumulated. Because an increase in systemic inflammatory markers can reflect progression of cancers, we hypothesized that temporal changes in NLR and PLR values can be used for a prediction model to detect cancer recurrence.

The prediction of cancer recurrence and the identification of risk factors have mainly been performed using logistic regression. With advances in ML techniques, many ML models for predicting cancer recurrence have recently been reported [32,33,34]. DL has been shown to enhance the performance and generalizability of ML models. DL can handle a huge amount of high-dimensional data and includes a feature selection algorithm [35,36,37]. Three representative DL models are deep neural networks (DNNs), convolution neural networks (CNNs), and recurrent neural networks (RNNs), and among them, RNNs, which have a loop structure and hidden states, have strength in handling sequential datasets [38]. In this study, it was difficult to apply RNN algorithms because the time series data collection interval was not constant, and the data collection frequency was different for each case. Therefore, we used DNN algorithms to extract the 12 features from time series blood laboratory data. In addition to this, RFE was used for feature selection. RFE, one of the most popular feature selection techniques, has shown excellent performances when applied with many ML and DL algorithms [19,39,40,41]. It recursively removes the weakest feature according to the ranks of the features. In this study, the DNN algorithm better predicted cancer recurrences than any of the traditional ML algorithms. When the RFE algorithm was applied to the DNN model, the ROC-AUC value was 0.883 ± 0.027 on the original training dataset, which was much higher than the ROC-AUC value of the LR (0.696 ± 0.059) with 16 features. All five selected features from the RFE algorithm were PLR-related features.

We also tried to adopt data augmentation by splitting the longitudinal blood test data for each patient. The performance of a DNN usually improves with the amount of training data. A large amount of data can also prevent overfitting problems or weak generalizability. Data augmentation is a technique that artificially creates new training data from existing training data. Various data augmentation techniques have been reported, especially for imaging data [42,43]. However, applying data augmentation on time series numerical data is difficult. The time between tumor initiation to overt presentation is not thought to take over six months. Therefore, we divided the time series data based on the 180 days before the diagnosis of recurrence for each recurrent case and created a new non-recurrent case with the data earlier than 180 days (Figure 1a). In this way, 117 non-recurrent cases were additionally generated from 206 recurrent cases. Overall, the number of data was augmented from 778 to 895 in the training dataset (Figure 1b). The cross-validation results showed comparable performances (0.889 ± 0.032) on the split (augmented) training dataset compared with those (0.883 ± 0.027) on the original training dataset. However, in external validation, the models trained with the augmented training dataset performed better than those trained with the original training dataset (Table 5). This means that data augmentation by data splitting can improve generalization performance. In addition, our model showed the ability to recognize longitudinal data samples correctly at each time point (Table 5).

In this study, we aimed to develop a DNN model to detect the recurrence after CCRT for head and neck cancers using post-treatment longitudinal laboratory data. The main limitation of this study is that the interval and amount of laboratory data are different for each case. If a larger amount of data can be obtained at regular intervals for a certain period, better ML and DL models can be developed. In addition, DNN models are often considered to be black boxes, making it challenging to understand the underlying decision-making process and resulting in a lack of interpretability. It can be difficult to explain how and why a model arrived at a particular prediction, limiting their interpretability and applications. While DNN models can learn complex representations from raw data, the learned features may not be easily interpretable or understandable by humans. This lack of transparency hinders the extraction of meaningful insights and an understanding of the underlying factors driving the models’ predictions. To address these limitations, various approaches, such as data augmentation, transfer learning, regularization methods, interpretability approaches, and adversarial training are needed to mitigate these challenges, and therefore enhance the performance and reliability of DNN models.

In this study, we extracted various features representing temporal changes in the NLR and PLR values in deep learning models and used them as input for training. Notably, applying RFE resulted in a performance of over 0.88 only with temporal changes in the PLR. With this, we interpreted the model as another follow-up tool for predicting recurrence and evaluated its value, which distinguishes our study from previous research that has mainly evaluated the predictive value of a single time point before or shortly after treatment. Furthermore, the application of RFE during the model construction process is also a significant result in terms of its technical aspect.

## 5. Conclusions

The DNN model with feature selection using the RFE algorithm showed the best performance among the ML models to discern a recurrence after CCRT in patients with head and neck cancers. The model performance could be enhanced with splitting and augmentation of the training dataset. The performance of the DNN-RFE model was validated with an external dataset. Systemic inflammatory markers can detect recurrence after treatment, and the predictive performance might be further improved with prospective data collection at regular intervals and periods.

## Figures and Tables

**Figure 1 cancers-15-03540-f001:**
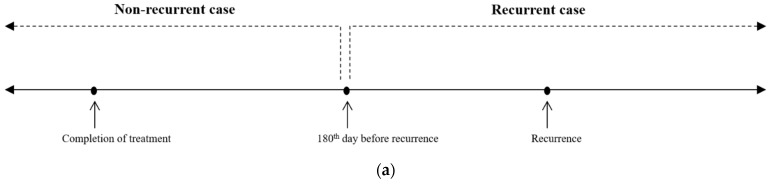
Sample split for recurrent cases. (**a**) For each recurrent case, the temporal blood laboratory values were split into two groups based on 180 days before the diagnosis of recurrence. (**b**) With data splitting for the recurrent cases, a modified dataset, named split training dataset, was built (*N* = 895).

**Figure 2 cancers-15-03540-f002:**
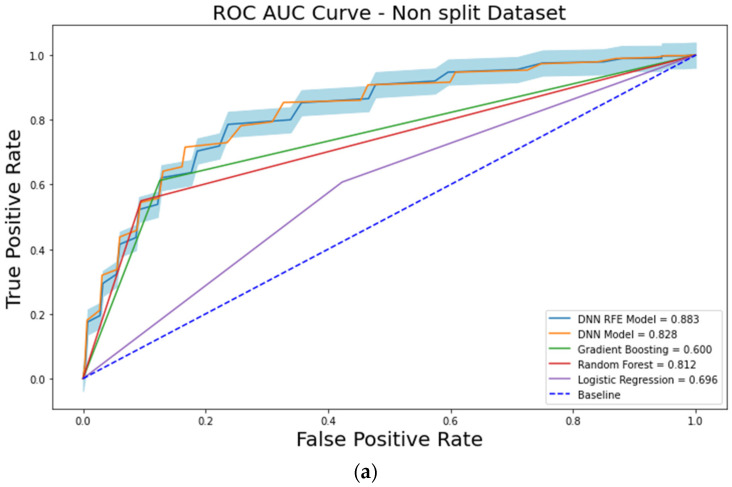
The receiver operating characteristic (ROC) curve analysis showing the performances of each machine learning model. (**a**) analysis results from the original dataset. (**b**) analysis results from the split dataset.

**Figure 3 cancers-15-03540-f003:**
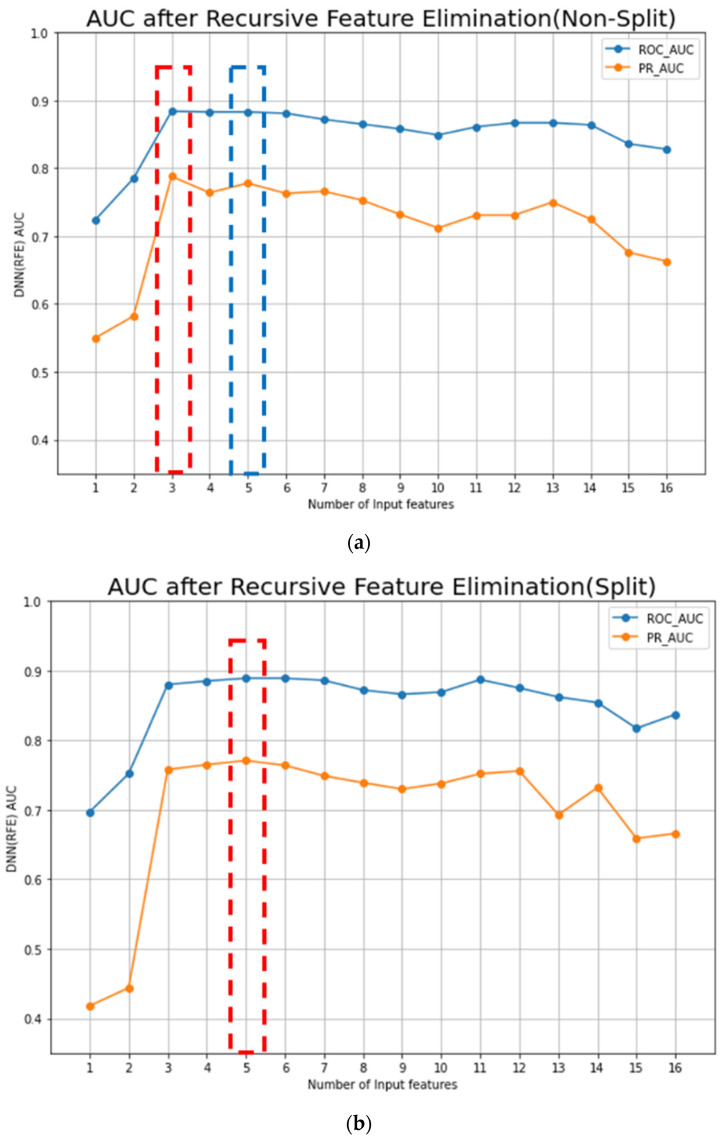
Performances of AI models from the original and split datasets: (**a**) The RFE algorithm is applied to the DNN model in the original (non-split) training dataset, and the top three features (PLR _mean_, PLR _Tx_, and PLR _max–min_) were selected. With these three features, the ROC-AUC and the PR-AUC values of the DNN-RFE model were 0.884 ± 0.032 and 0.788 ± 0.032, respectively, in the original training dataset; (**b**) the DNN-RFE algorithm is applied in a split training dataset, and the PLR _R_, PLR _R-Tx_, PLR _mean_, PLR _Tx_, and PLR _max–min_ were selected. With the top five features, the ROC-AUC and the PR-AUC values of the DNN-RFE model were 0.889 ± 0.032 and 0.771 ± 0.044, respectively, in the split training dataset. When these top five features were used in the original training dataset, the ROC-AUC was 0.883 ± 0.027 (Table 3(a)).

**Table 1 cancers-15-03540-t001:** The definition of 12 blood laboratory meta-features.

Features	Definition
NLR _mean_, PLR _mean_	Mean NLR (PLR)
NLR _max–min_, PLR _max–min_	Maximum NLR (PLR) *minus* minimum NLR (PLR)
NLR _Tx_, PLR _Tx_	NLR (PLR) after treatment *
NLR _R_, PLR _R_	NLR (PLR) at the time of recurrence or at the end of the period **
NLR _R-mean_, PLR _R-mean_	NLR _R_ (PLR _R_) *minus* NLR _mean_ (PLR _mean_)
NLR _R-Tx_, PLR _R-Tx_	NLR _R_ (PLR _R_) *minus* NLR _Tx_ (PLR _Tx_)

* NLR _Tx_ (PLR _Tx_): NLR (PLR) after treatment (values from 14 days to 90 days after the end of treatment). ** NLR _R_ (PLR _R_): NLR (PLR) at the time of recurrence for recurrent cases and at the end of the period for non-recurrent cases.

**Table 2 cancers-15-03540-t002:** Study patients (*N* = 778).

Characteristics		Values
Age (year)	Mean ± SD	60.4 ± 13.0
Gender (male/female)		625/153
Smoking		481 (61.8%)
Stage *	I	11 (1.4%)
	II	68 (8.8%)
	III	209 (26.9%)
	IV	489 (62.9%)
Tumor sites	Oral cavity	45 (5.8%)
	Larynx	54 (6.9%)
	Oropharynx	228 (29.3%)
	Hypopharynx	74 (9.5%)
	Nasopharynx	296 (38.0%)
	PNS/nasal cavity	42 (5.4%)
	Others	39 (5.0%)
Pathology	SCC	503 (64.7%)
	UDC	243 (31.2%)
	Others	32 (4.1%)
Follow-up period (months)	Median (interquartile range)	41.8 (17.4~71.6)
Recurrences		206 (26.5%)
RFS (months)	Median (SE, 95% CI)	189.8 (33.5, 124.2~255.3)
5-year RFS rate		66.5%

* Staging according to the *AJCC* 7th edition. The value for one case was missing. SD, standard deviation; SCC, squamous cell carcinoma; UDC, undifferentiated carcinoma; RFS, relapse-free survival; SE, standard error.

**Table 3 cancers-15-03540-t003:** The performance of the trained models.

**a. from Original Dataset**							
**Model**	**Training**	**Original Dataset**					
	**Features (N)**	**ROC-AUC**	**PR-AUC**	**Accuracy**	**Precision**	**Sensitivity**	**Specificity**
Logistic regression	16	0.696 ± 0.059	0.454 ± 0.079	0.770 ± 0.039	0.606 ± 0.105	0.885 ± 0.043	0.461 ±0.108
Random forest	16	0.812 ± 0.049	0.622 ± 0.081	0.855 ± 0.027	0.758 ± 0.058	0.917 ± 0.026	0.687 ± 0.081
Gradient boosting	16	0.600 ± 0.052	0.346 ± 0.055	0.825 ± 0.043	0.744 ± 0.116	0.926 ± 0.038	0.554 ± 0.065
DNN	16	0.828 ± 0.032	0.663 ± 0.069	0.686 ± 0.102	0.471 ± 0.112	0.675 ± 0.070	0.715 ± 0.070
DNN-RFE	5	0.883 ± 0.027	0.778 ± 0.042	0.801 ± 0.030	0.600 ± 0.041	0.800 ± 0.053	0.801 ± 0.053
**b. from Split Dataset**							
**Model**	**Training**	**Split Dataset**					
	**Features (N)**	**ROC-AUC**	**PR-AUC**	**Accuracy**	**Precision**	**Sensitivity**	**Specificity**
Logistic regression	16	0.700 ± 0.053	0.456 ± 0.072	0.759 ± 0.027	0.482 ± 0.200	0.951 ± 0.029	0.139 ± 0.070
Random forest	16	0.751 ± 0.053	0.539 ± 0.084	0.845 ± 0.028	0.737 ± 0.084	0.940 ± 0.023	0.536 ± 0.082
Gradient boosting	16	0.735 ± 0.057	0.516 ± 0.085	0.822 ± 0.046	0.731 ± 0.114	0.922 ± 0.038	0.554 ± 0.079
DNN	16	0.837 ± 0.038	0.666 ± 0.045	0.786 ± 0.026	0.537 ± 0.044	0.808 ± 0.029	0.715 ± 0.029
DNN-RFE	5	0.889 ± 0.032	0.771 ± 0.044	0.812 ± 0.014	0.578 ± 0.022	0.827 ± 0.094	0.763 ± 0.094

DNN, deep neural network; DNN-RFE, deep neural network-recursive feature elimination; ROC-AUC, receiver operating curve-area under the curve; PR-AUC, precision recall-area under the curve.

**Table 4 cancers-15-03540-t004:** Feature selection of the DNN model using the RFE algorithm.

No. of Features	Eliminated Feature	Original Training Dataset		Split Training Dataset	
		ROC-AUC	PR-AUC	ROC-AUC	PR-AUC
16	Sex	0.828	0.663	0.837	0.666
15	Age	0.836	0.676	0.817	0.659
14	Smoking	0.864	0.725	0.854	0.732
13	Stage	0.867	0.750	0.862	0.693
12	NLR _max–min_	0.867	0.731	0.875	0.756
11	NLR _R_	0.861	0.731	0.887	0.752
10	NLR _Tx_	0.849	0.712	0.869	0.738
9	NLR _mean_	0.858	0.732	0.866	0.730
8	NLR _R-mean_	0.865	0.753	0.872	0.739
7	NLR _R-Tx_	0.872	0.766	0.886	0.749
6	PLR _R-mean_	0.881	0.763	0.889	0.764
5	PLR _R_	0.883	0.778	0.889	0.771
4	PLR _R-Tx_	0.883	0.764	0.885	0.765
3	PLR _mean_	0.884	0.788	0.880	0.758
2	PLR _Tx_	0.785	0.582	0.752	0.444
1	PLR _max–min_	0.724	0.550	0.697	0.418

DNN, deep neural network; RFE, recursive feature elimination; ROC-AUC, receiver operating curve-area under the curve; PR-AUC, precision recall-area under the curve.

**Table 5 cancers-15-03540-t005:** External validation and cross-validation performance.

Training Dataset		Split Training		Original Training		Split Training		Original Training	
Validation Dataset		Split Validation		Split Validation		Original Validation		Original Validation	
Model	Features	ROC-AUC	PR-AUC	ROC-AUC	PR-AUC	ROC-AUC	PR-AUC	ROC-AUC	PR-AUC
LR	16	0.571	0.404	0.582	0.406	0.556	0.481	0.572	0.494
RF	16	0.531	0.352	0.538	0.356	0.531	0.449	0.748	0.702
GB	16	0.717	0.550	0.595	0.412	0.672	0.596	0.585	0.496
DNN	16	0.832	0.737	0.558	0.395	0.766	0.766	0.643	0.560
DNN-RFE	5	0.845	0.785	0.708	0.545	0.784	0.723	0.710	0.670

LR, logistic regression; RF, random forest; GB, gradient boosting; DNN, deep neural network; DNN-RFE, deep neural network-recursive feature elimination.

## Data Availability

The data presented in this study are available on request from the corresponding author. The data are not publicly available because there was no pre-approval by IRB.

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
