# Peer review of "Detection of Cancer Recurrence Using Systemic Inflammatory Markers and Machine Learning after Concurrent Chemoradiotherapy for Head and Neck Cancers"

_cancers, 2023, doi:10.3390/cancers15143540_

Round 1

Reviewer 1 Report

The authors aimed to predict cancer recurrence with inflammatory markers and machine learning techniques. The EMR data in this study was collected from 1994 to 2019. The features used in this study including, age, sex, smoking, stage, and 12 blood lab features. The 12 blood features were performed at Table 1 as meta-features. Next, the procedure of data splitting was presented at Figure 1. Finally, the performance of models were presented at Figure 2 and Table 3. The best AUC they received was from deep neural network with recursive feature elimination.

The manuscript is a well-presented article. However, some questions are remained.

Introduction

1.      The author should introduce the relationship between NLP, PLR, and head and neck cancers more.

2.      The author should introduce feature engineering such as meta-features and recursive feature elimination.

Experimental design and presentation

1.      The title has mentioned about systemic inflammation, why did not include other related markers?

2.      The study collected data from 1994 to 2019. The data format, laboratory tests machine must be changed during the time. The author should clarify how did they handle the data across about 25 years.

3.      Since this is a sort of retrospective study, I assume the days when patients were found recurrence, 180th day before recurrence, and the day patients’ blood were tested are different. This may cause the blood sample collected from patients were in different status. The author should clarify this issue.

4.      How did the authors decide to create such meta-features? (if it will be mentioned at introduction, then I think it’s okay to keep it simple here)

5.      The authors must present the accuracy, precision, sensitivity, and specificity while presenting the performance of models.

Author Response

I appreciate you for the thorough and excellent review of this paper. Below, I have provided detailed responses to each comment, along with the corresponding revisions made to the main text.   

Introduction

1. The author should introduce the relationship between NLP, PLR, and head and neck cancers more.

-> Thank you for this comment. We add some explanation about the the relationship between NLP, PLR, and head and neck cancers in the introduction (line 69-75) as below.

“Neutrophilia and thrombocytosis have been shown to reflect tumor progression. On the other hand, lymphopenia, which indicates a compromised anti-tumor immune response, has been associated with a poor prognosis in various advanced solid tumors. The neu-trophil-to-lymphocyte ratio (NLR) and platelet-to-lymphocyte ratio (PLR) may capture the combined effects of these opposing factors. High NLR and PLR have been widely recognized as poor prognostic indicators for recurrence and survival in head and neck cancer.” 

2. The author should introduce feature engineering such as meta-features and recursive feature elimination.

 -> Thank you for the comment. As you suggested, we added the following paragraph in section 2.2 (line 135-144 in Methods).

 Missing values were removed, and categorical variables were encoded using the one-hot encoding technique since there was no ordinal relationship. Subsequently, the 12 extracted input variables were normalized to a standard scale. Additionally, recursive feature elimination was performed to assess the impact of each input variable on the response variable. The contribution of each variable to the prediction was calculated as a discrete interval encompassing all the variables in the training dataset. The response probability was estimated for each range of the input variable, while keeping the other variables at their average value for numeric variables or at zero for binary variables. In each iteration, the input variable with the smallest dynamic range of the output response was successively removed until only one variable remained.

Experimental design and presentation

  1. The title has mentioned about systemic inflammation, why did not include other related markers?

-> Thank you for your comment. First, this is a retrospective study on the data over a very long time span. CBC (Complete Blood Count) is the blood test that has been most consistently measured in such a large number of patients over a long period of time. With CBC, we can calculate NLR and PLR easily. NLR (Neutrophil-to-Lymphocyte Ratio) and PLR (Platelet-to-Lymphocyte Ratio) can reflect two opposing streams of systemic inflammatory markers, such as neutrophils and platelets vs. lymphocytes. These two markers are recognized as representative markers, and we aimed to avoid making the model overly complex by including numerous markers.

  1. The study collected data from 1994 to 2019. The data format, laboratory tests machine must be changed during the time. The author should clarify how did they handle the data across about 25 years.

-> Thank you for the comment. We’d like you to understand the process of assessing equivalency for blood lab test in this institution as below.

As a traditional diagnostic test, the complete blood count (CBC) test has not undergone any significant change in its data format so far. In addition, since the consistency of CBC test results is important, whenever the test equipment was changed within the institution, a comparison evaluation was conducted to verify the equivalence between the new and previous test equipment. Through this process, the test results were maintained consistently and reliably.

  1. Since this is a sort of retrospective study, I assume the days when patients were found recurrence, 180thday before recurrence, and the day patients’ blood were tested are different. This may cause the blood sample collected from patients were in different status. The author should clarify this issue.

   ->  First, we apologize for the confusion. Blood labs were repeatedly conducted on each patient during the follow-up after the end of treatment. All of these labs were utilized, rather than relying on a single-point lab result. In cases where recurrence occurred during the follow-up period, we assumed that blood tests conducted prior to the 180th day before recurrence do not have any association with recurrence and do not reflect its occurrence. Therefore, we split the data based on the 180th day before recurrence for individual cases of recurrence.

  1. How did the authors decide to create such meta-features? (if it will be mentioned at introduction, then I think it’s okay to keep it simple here)

-> Thank you for the comment. As you know, recent deep learning technologies have the capability to capture the temporal patterns of time-series data. However, it is often very difficult to learn such a pattern if there are a limited amount of data and also they are acquired irregularly for each patient. We actually tried to use recurrent neural networks (RNN) to do so, but failed to train a decent prediction model. And we think it might be because of the amount of training data and the data irregularity. This is the main reason why we come up with those meta-features.

So, we revised the manuscript in section 2.2. as follows (line 120-123).

Recent deep learning technologies have the capability to capture the temporal patterns of time-series data. However, it is often very difficult to learn such a pattern if there are a limited amount of data and they are acquired irregularly for each patient.

  1. The authors must present the accuracy, precision, sensitivity, and specificity while presenting the performance of models.

    -> Thank you for the comment. We added data on the accuracy, precision, sensitivity, and specificity of the performance (Table 3).

Reviewer 2 Report

The paper reports the study of development of DNN model from the temporal NLR and PLR values during follow-up after CCRT and to evaluate the model performance compared with the conventional ML models. The reviewer thinks this article is very interesting for cancer research, however, it needs to clear and complete in this article. To render the manuscript suitable for publication to Cancers, several corrections should be made before the paper should be accepted.

Specific Comments:

·       The quality of the figures 2,3 needs to be improved.

·       In this research, please add more discussion about the effect of stage, tumor sites and pathology with NLR and PLR values after CCRT in head and neck cancer patients in the manuscript. It would be helpful if the authors give example or scenario to support its description. Clarification of this point in text is needed.

·       Please discuss more detail about the limitation of DNN models in the manuscript. In addition, the authors need to highlight the new findings and their connection with previous knowledge.

Minor editing of English language required.

Author Response

I appreciate you for the thorough and excellent review of this paper. Below, I have provided detailed responses to each comment, along with the corresponding revisions made to the main text. 

Specific Comments:

  1. The quality of the figures 2,3 needs to be improved.

The figures were revised and reattached to the manuscript.

  1. In this research, please add more discussion about the effect of stage, tumor sites and pathology with NLR and PLR values after CCRT in head and neck cancer patients in the manuscript. It would be helpful if the authors give example or scenario to support its description. Clarification of this point in text is needed.

Thank you for your brilliant comment.

We added some sentences and make some changes to several sentences to clarify this issue (line 338 – 348 in Discussion) like below. Changes were highlighted within the manuscript.

“Prominent prognostic factors for cancer recurrence include stage, pathology, and site. Tumors diagnosed at an advanced stage, those exhibiting aggressive pathology with poor differentiation, or those located in sites such as the oral tongue, are associated with a poor prognosis. In addition to these factors, patients with higher NLR or PLR at diagnosis are now believed to have an increased likelihood of experiencing recurrence after treatment.

Unlike previous studies, we employed serially measured NLRs and PLRs after the completion of treatment, rather than the point values at diagnosis. Our objective was not to assess the prognostic value of those systemic inflammatory markers. Instead, we aimed to determine whether the temporal values measured after treatment could indicate the presence of recurrence, if it occurs.”

  1. Please discuss more detail about the limitation of DNN models in the manuscript. In addition, the authors need to highlight the new findings and their connection with previous knowledge.

Thank you for your excellent comment. We added paragraphs in the Discussion section as below (line 398-416 in Discussion). Also, DNN models are often considered black boxes, making it challenging to understand the underlying decision-making process and resulting in a lack of interpretability. It can be difficult to explain how and why the model arrived at a particular prediction, limiting their interpretability and applications. While DNN models can learn complex representations from raw data, the learned features may not be easily interpretable or understandable by humans. This lack of transparency hinders the extraction of meaningful insights and an understanding of the underlying factors driving the model's predictions. To address these limitations, various approaches, such as data augmentation, transfer learning, regularization methods, interpretability approaches, and adversarial training, are needed to mitigate these challenges and enhance the performance and reliability of DNN models.

In this study, we extracted various features representing the temporal changes of NLR and PLR in deep learning models and used them as input for training. Notably, applying RFE resulted in a performance of over 0.88 only with the temporal changes of PLR. With this, we interpreted the model as another follow-up tool for predicting recurrence and evaluated its value, which distinguishes our study from previous research that mainly evaluated the predictive value of a single time point before or shortly after treatment. Furthermore, the application of RFE during the model construction process is also a significant result in terms of its technical aspect.

Reviewer 3 Report

The authors did nice research work and they have presented interesting findings which are not totaly new but still has some scientific value. In order to improve curent research paper authors must update there paper with the answers to questions given below:

1. Authors in the chapter 2.1 state that they exclude patients that are smoking, but in chapter 2.2 and others with results smoking factor is used as input feature. Please fix that.

2. Augmentation is always difficult when you have limited time series data, but data spliting itself is not real augmentation it is just a partioning of available data. Did you try to add some small noise to each  feature input?

3. How authors deals with unbalanced data. Clearly the number of recurrence cases is highe that non-recurrence. Please explain.

4. All mentioned decision making algorithms and models has limitations and advantages, but from the given text is not clear how complex models was. Please add brief describtion about each tested decision making models. Readers would be interested in DNN structure, hyperparameters, RF structure and other parameters that are important in understanding of model complexity.

5. It is not clear how many times same pantient were measured. How many time points were taken and how many time points are needed for accurate forecasting?

Author Response

I appreciate you for the thorough and excellent review of this paper. Below, I have provided detailed responses to each comment, along with the corresponding revisions made to the main text. 

  1. Authors in the chapter 2.1 state that they exclude patients that are smoking, but in chapter 2.2 and others with results smoking factor is used as input feature. Please fix that.

Thank you for your comment, but it seems like there’s a kind of misunderstanding. In chapter 2.1, we described, ‘We also excluded the patients with missing analysis items; missing blood test data, missing medical records on the recurrences, follow-up status, or smoking.’. We meant that we excluded the patients with missing medical records on smoking, not those with smoking. I ask for your understanding.

  1. Augmentation is always difficult when you have limited time series data, but data spliting itself is not real augmentation it is just a partioning of available data. Did you try to add some small noise to each feature input?

Thank you for your valuable comment. We fully understand your concern. Data augmentation techniques such as synthetic decimal oversampling (SMOTE) and Gaussian noise upsampling can help the model better learn unbalanced data, increase the randomness of the data, and improve the model's generalization performance.

Nevertheless, in some cases, augmentation techniques can distort the distribution of the original data, and the addition of such noise to the data also risks the model learning the wrong pattern and reducing the reliability of the prediction results. We also tried to use such augmentation techniques, but failed to get better results than when we used the original dataset.

As in our results, our data partitioning approach did not degrade the prediction performance compared to the ones with the original dataset in the internal cross validation study and showed slightly higher performance in the external validation. Moreover, by splitting the data, we showed that our model could recognize longitudinal data samples correctly at each time point, as mentioned in the Discussion section.

Thus, we hope that you understand that it is worth introducing our data portioning approach for the readers as a simple data augmentation and also as a way to see if the model can correctly predict the longitudinal data at each time point.

  1. How authors deals with unbalanced data. Clearly the number of recurrence cases is higher that non-recurrence. Please explain.

Thank you for your comment, but I ask you to see Figure 1b as the reference on the patient population. As you see in Figure 1b, there were 206 recurrent cases and 572 non-recurrent cases before data splitting. For 206 recurrent cases, we split the temporal lab data based on the 180th day before the recurrence so that data before the day were reclassified as non-recurrent for each patient. With this data split, 117 non-recurrent data were generated and added. Resultantly, we had 206 recurrent cases and 689 non-recurrent cases after data splitting, in which the number of non-recurrent cases was still higher than the number of recurrent cases.

  1. All mentioned decision making algorithms and models has limitations and advantages, but from the given text is not clear how complex models was. Please add brief describtion about each tested decision making models. Readers would be interested in DNN structure, hyperparameters, RF structure and other parameters that are important in understanding of model complexity.

Thank you for your comment. We added the following contents in the section 2.4 (line 165-188).

Logistic regression is a statistical model used for binary classification problems. It estimates the probability of an instance belonging to a particular class by utilizing a logistic function. This model assumes a linear relationship between the input features and the log-odds of the target class. However, it may struggle to capture complex nonlinear relationships and may not perform as effectively as more advanced models in highly complex tasks.

Random forest, on the other hand, is an ensemble learning method that combines multiple decision trees to make predictions. It creates a forest of decision trees, where each tree is trained on a random subset of the data and features. Random forest is capable of performing both classification and regression tasks and can handle complex interactions and nonlinear relationships in the data. However, it can be prone to overfitting if the number of trees is too high and may not perform as well on high-dimensional datasets.

Gradient boosting is another ensemble learning technique that builds an ensemble of weak prediction models, typically decision trees, in a sequential manner. Gradient boosting combines the predictions of multiple weak models to make a final prediction. It is capable of handling complex relationships, capturing interactions, and performing well on structured and tabular data. However, it can be computationally expensive and requires careful tuning of hyperparameters.

In contrast to logistic regression, which assumes a linear relationship between the input features and the log-odds of the target class, Deep Neural Networks (DNNs) can approximate any complex function by learning hierarchical representations through hidden layers. This flexibility enables DNNs to handle highly nonlinear data and capture intricate interactions and dependencies between features. DNNs excel in tasks where the relationships in the data are complex and may not be easily captured by simpler models.

  1. It is not clear how many times same patient were measured. How many time points were taken and how many time points are needed for accurate forecasting?

Thank you for your great comment. Ideally, it would be good to know the number of time points or intervals of measurement needed for accurate forecasting. However, this is a retrospective study of the data over a very long time span. So, the number of time points or the intervals of measurement vary widely. This is also why we did not build an RNN model for this data. We described this as a limitation of the study in the last part of the discussion (lines 395-396). It is partially because follow-up durations vary for each patient and because blood labs were more frequently taken for those with several health issues. the number of taking blood labs ranged from 3 to 126. As you mentioned, we’re planning a prospective study in which a fixed interval of blood lab measurements will be applied to included patients to know the ideal number and interval of blood lab.

Round 2

Reviewer 1 Report

The questions have been answered thoroughly and the manuscript has been improved dramatically.